# Restoration of the Phenotype of Dedifferentiated Rabbit Chondrocytes by Sesquiterpene Farnesol

**DOI:** 10.3390/pharmaceutics14010186

**Published:** 2022-01-13

**Authors:** Guan-Xuan Wu, Chun-Yu Chen, Chun-Shien Wu, Lain-Chyr Hwang, Shan-Wei Yang, Shyh-Ming Kuo

**Affiliations:** 1Department of Biomedical Engineering, I-Shou University, Kaohsiung City 82445, Taiwan; turtle01311995@gmail.com (G.-X.W.); iergy2000@gmail.com (C.-Y.C.); 2Orthopedics Department, Kaohsiung Veterans General Hospital, Kaohsiung City 84001, Taiwan; 3Center of General Education, I-Shou University, Kaohsiung City 82445, Taiwan; wucs@isu.edu.tw; 4Department of Electrical Engineering, I-Shou University, Kaohsiung City 82445, Taiwan; lain@isu.edu.tw

**Keywords:** sesquiterpene farnesol, collagen, osteoarthritis, dedifferentiation, glycosaminoglycan

## Abstract

Osteoarthritis (OA) is a joint disorder characterized by the progressive degeneration of articular cartilage. The phenotype and metabolism behavior of chondrocytes plays crucial roles in maintaining articular cartilage function. Chondrocytes dedifferentiate and lose their cartilage phenotype after successive subcultures or inflammation and synthesize collagen I and X (COL I and COL X). Farnesol, a sesquiterpene compound, has an anti-inflammatory effect and promotes collagen synthesis. However, its potent restoration effects on differentiated chondrocytes have seldom been evaluated. The presented study investigated farnesol’s effect on phenotype restoration by examining collagen and glycosaminoglycan (GAG) synthesis from dedifferentiated chondrocytes. The results indicated that chondrocytes gradually dedifferentiated through cellular morphology change, reduced expressions of COL II and SOX9, increased the expression of COL X and diminished GAG synthesis during four passages of subcultures. Pure farnesol and hyaluronan-encapsulated farnesol nanoparticles promote COL II synthesis. GAG synthesis significantly increased 2.5-fold after a farnesol treatment of dedifferentiated chondrocytes, indicating the restoration of chondrocyte functions. In addition, farnesol drastically increased the synthesis of COL II (2.5-fold) and GAG (15-fold) on interleukin-1β-induced dedifferentiated chondrocytes. A significant reduction of COL I, COL X and proinflammatory cytokine prostaglandin E2 was observed. In summary, farnesol may serve as a therapeutic agent in OA treatment.

## 1. Introduction

Articular cartilage has a limited capacity to heal because of its lack of blood vessels and lymphatics. Minor injury to articular cartilage gradually results in progressive damage and degenerative diseases, such as osteoarthritis (OA), if not appropriately treated [1,2]. Chondrocytes are responsible for the synthesis, maintenance and degradation of the extracellular matrix (ECM) in response to signals from cytokines, inflammatory mediators and matrix fragments [3,4]. However, the low cell density of chondrocytes in the cartilage (1–5% of the total volume) and avascular nature of the tissue limit the potential for regeneration and repair following cartilage injury. Studies have demonstrated that the phenotypic instability of chondrocytes or changes in chondrocyte morphology and volume occurring before marked cartilage degeneration might be a crucial indicator of, or a major feature in, OA development [5]. Articular chondrocytes in a healthy condition are typically quiescent and highly differentiated cells, which maintain the ECM. These phenotypically stable chondrocytes normally synthesize a tough, basket-weave matrix principally composed of COL II and aggrecan. The maintenance of this chondrocytic phenotype to produce a resilient native hyaline-like matrix is crucial in tissue engineering strategies [6,7].

To date, many cell-based therapies have been developed to treat cartilage damage or OA, including autologous chondrocyte implantation (ACI), with cells kept in place either by using a periosteal flap or seeding cells into a cell sheet matrix to postpone or avoid knee replacement surgery at the end-stage of cartilage degeneration [8,9]. For the ACI technique, the chondrocytes expanded ex vivo lose their phenotype and the ability to synthesize COL II and aggrecan. This is due to the long expansion time and multiple passages in monolayer cultures needed to obtain workable amounts of chondrocytes in the in vitro environment. As dedifferentiation progresses, the synthesis of ECM molecules, such as COL II and glycosaminoglycans (GAGs), is lost, whereas chondrocytes that produce different extracellular proteins (e.g., collagen types I and X (COL I and COL X)) incorporate into the matrix, providing inferior mechanical properties [10,11]. Therefore, although ACI yields impressive clinical results in repairing damaged cartilage, a more efficient method, particularly one that prevents chondrocyte dedifferentiation, is clearly required.

Many investigations have reported techniques for decelerating or preventing dedifferentiation in cartilage therapy through the supplementation of the culture medium with different growth factors [12,13] or the use of three-dimensional (3D) cultures that favor chondrocyte phenotype maintenance to re-differentiate the differentiated chondrocytes [14]. Generally, numerous changes occur in dedifferentiated chondrocytes, including distorted cell morphology, a decreased expression of the chondrogenic transcription factor SOX9 and suppressed production of cartilage-specific matrix molecules COL II and aggrecan. However, the synthesis of COL I and COL X is markedly increased [15]. Although the 3D culture system (rather than 2D approaches) is used exclusively to maintain the chondrocyte phenotype, the difficulties involved in the readout or assay of cell behaviors, such as protein expressions, hamper quantification. Therefore, some investigations still use 2D culture flasks to explore the cellular response pretreatment and posttreatment with some chemicals.

Interleukin (IL)-1β is a pleiotropic proinflammatory cytokine that mediates cartilage degeneration in osteoarticular disorders. IL-1β can activate matrix-degrading enzymes, regulate the expression of matrix components and induce chondrocyte apoptosis [16]. Furthermore, this cytokine exacerbates local tissue inflammation through the stimulation of cells to upregulate proinflammatory cytokines and then activates the expression of proteolytic enzymes such as matrix metalloproteinases (MMP) in cells, which leads to cartilage matrix degradation. In osteoarthritis, the proinflammatory factors such as interleukin-1β and tumor necrosis factor α activate NF-κB signals. Afterward, the chondrocytes undergo different phenotypic changes in response to inflammation, further increasing the expression of TNF-α, IL-1β, IL-6 and MMPs, leading to chondrocyte apoptosis. Moreover, NF-κB also activates iNO, PGE2 and COX-2 to promote chondrocyte catabolism.

Farnesol, a naturally occurring sesquiterpene compound, has shown anti-inflammatory, antioxidant and antimicrobial characteristics [17]. Studies have demonstrated that farnesol downregulates essential inflammatory cytokines, such as IL-1, IL-6 and tumor necrosis factor alpha (TNF-α), in vivo [18,19]. Our previous study indicated that farnesol significantly improved wound healing and rotator cuff repair through the reduction of oxidative stress and inflammation and a significant enhancement of collagen production [20,21]. These findings suggest that farnesol can modulate connective tissue and ECM synthesis and promote wound healing in tissue engineering applications.

Taken together, chondrocytes can be dedifferentiated either with multiple passages through subcultures in a 2D culture flask or through IL-1β induction, leading to morphological changes and a loss of capability to synthesize COL II and GAG. The goal of this study was to investigate the restoration effects of farnesol on dedifferentiated chondrocytes by assessing the chondrogenic expressions of SOX9, COL II and GAG and dedifferentiation-related protein COL I, COL X and PGE2 production by using immunofluorescence staining, Western blotting and collagen assay kits. Moreover, the farnesol-encapsulated hyaluronan (Farn/HA) nanoparticle was prepared and its restoration effects evaluated with pure farnesol.

## 2. Materials and Methods

### 2.1. Materials

Farnesol, hyaluronan (HA, MW: 9 × 10^5^ Da), 3-4,5-dimethylthiazol-2-yl-2,5-diphenyltetrazolium bromide (MTT) and FeCl_3_ were purchased from Sigma (St. Louis, MO, USA) (Version 1.50; National Institutes of Health, Bethesda, MD, USA). F12 medium, fetal bovine serum, trypsin, streptomycin and penicillin were obtained from Gibco (Waltham, MA, USA). All other chemicals used in this study were of analytical reagent grade and used without further purification. Animal experiments conducted in this study were approved by the Institutional Animal Care and Use Committee of I-Shou University, Kaohsiung, Taiwan (IACUC-ISU-108-035, approval date: 12 February 2020).

### 2.2. Isolation and Culture of Rabbit Chondrocytes

Chondrocytes were isolated from 3-week-old New Zealand rabbits. After the animals were sacrificed with cardiac injections of saturated potassium chloride, the articular cartilage was collected using a surgical blade and immersed in a phosphate-buffered saline solution. The tissues and ligaments were cleaned off before the cartilage fragments were sliced, soaked in protease (2 mg/mL) for 2 h and subsequently dissociated in serum-free F12 media containing COL II (2 mg/mL) for 20 h. The cell suspension was centrifuged at 1200 rpm for 5 min to remove the upper layer of the supernatant; subsequently, the chondrocytes were cultured in F12 media containing 10% bovine serum and antibiotics (penicillin/streptomycin, 200 U/mL) and incubated at 37 °C in 5% CO_2_. On confluence, the chondrocytes were trypsinized, and half of the cells were replated (passaged cultures).

### 2.3. Production and Characterization of Farn/HA Nanoparticle

A Farn/HA nanoparticle was fabricated using an electrostatic field system according to our previous reports [22]. Briefly, a stock solution of 1.2-M farnesol was prepared through the dissolution of farnesol in dimethyl sulfoxide (DMSO). Subsequently, 1 mL of HA solution (0.2 mg/mL) was carefully mixed with 30 μL of the farnesol solution. The mixture was transferred onto a Petri dish and placed between two plate electrodes. The preparation parameters of the Farn/HA nanoparticle in the electrostatic field system were as follows: applied electric field strength: 5.0 kV/cm, reaction time: 1 h, crosslinking reagent: 0.001-N FeCl_3_ and environmental temperature: 17 °C. The prepared Farn/HA nanoparticles were initially placed on formvar-coated copper grids. These grids were negatively stained with 2% phosphotungstic acid and air-dried. The air-dried grids were examined and analyzed using a Tecnai G2 20 S-Twin transmission electron microscope (TEM; FEI, Hillsboro, OR, USA). The sizes of the Farn/HA nanoparticles were estimated through TEM by randomly sampling approximately 60 individual Farn/HA nanoparticles, which minimized any selection bias.

The encapsulation efficiency (EE) of farnesol in the Farn/HA nanoparticle was determined as follows: 1 mL of solution containing the prepared Farn/HA nanoparticle was added to a centrifuge tube and then centrifuged at 10,000 rpm for 10 min. Subsequently, the amount of nonencapsulated farnesol in the supernatant was determined through high-performance liquid chromatography (HPLC, Agilent 1100 series, Santa Clara, CA, USA). The EE of farnesol in the Farn/HA nanoparticle was calculated using the following formula:EE (%) = [(total amount farnesol − amount of nonencapsulated farnesol)/total amount of farnesol)] × 100%(1)

The in vitro release of farnesol from the Farn/HA nanoparticle was investigated through the addition of 1 mL of Farn/HA nanoparticle solution into a 1.5-mL microcentrifuge tube. The tube was subsequently placed on a 40 rpm shaker at 37 °C. At defined time points, the sample was centrifuged at 10,000 rpm for 10 min. The amount of released farnesol in the supernatant was determined through high-performance liquid chromatography. The experiments were performed in triplicate.
In vitro release (%) = [(total amount of farnesol − residue of farnesol)/total amount of farnesol] × 100%(2)

### 2.4. Cell Viability

The effects of farnesol and the Farn/HA nanoparticle on chondrocyte viability were assessed using the MTT assay. The chondrocytes (passage 2, 7 × 10^3^ cells/well) were seeded onto 96-well plates for 24 h and subsequently treated with various concentrations of farnesol or Farn/HA nanoparticles in triplicate. After 24 h of treatment exposure, 20 μL of the MTT solution (5 mg/mL) was added to each well, and the cells were incubated for an additional 3 h. The formed formazan precipitate was dissolved in 200 μL of DMSO, and the solution was vigorously mixed to dissolve the dye. The absorbance was measured at 570 nm by using a multiplate reader (Thermo Scientific, Waltham, MA, USA). ImageJ software was used to measure the cell area (three images).

### 2.5. Collagen Quantification and Measurement of PGE2

COL I and II synthesis was quantified with enzyme-linked immunoassay (ELISA) by using type I and II collagen detection kits following the manufacturer’s protocols (Chondrex Inc., Woodinville, WA, USA). Briefly, chondrocytes (passage 2, 1.2 × 10^5^ cells/well) were seeded in a six-well plate for 24 h and then treated with 10-ng/mL IL-1β for 24 h. This was followed by treatment with farnesol and Farn/HA nanoparticles for 1, 3 and 5 d. Thereafter, the medium was collected for the COL I and II and PGE2 assays by using ELISA kits. The absorbance for COL I and II analysis was read at 490 nm, and the PGE2 level was read at 450 nm by using a microplate reader (Thermo Scientific, Waltham, MA, USA).

### 2.6. Quantification GAG Assay

The content of GAG in the chondrocytes pretreatment and posttreatment with farnesol and Farn/HA nanoparticles was measured using a Blyscan Glycosaminoglycan Assay kit (Biocolor, Antrim, UK). Briefly, samples were digested with papain extraction buffer (100 μg/mL in Na_2_HPO_4_-NaH_2_PO_4_ buffer, pH 6.4) for 3 h at 65 °C. Subsequently, the digested extracts were removed and centrifuged at 10,000 rpm for 10 min. The supernatants were collected and dissolved in the dissociation reagent. Finally, the absorbance of the blanks, standards and test samples was measured at 656 nm with a microplate reader.

### 2.7. Immunofluorescence Staining

Chondrocytes with and without treatment were fixed with 4% paraformaldehyde for 10 min, treated with 0.1 Triton X-100 for 10 min and blocked with 5% fetal bovine serum (FBS) for an additional 60 min. Chondrocytes were incubated overnight with primary antibodies against COL II (1:500), COL I (1:500) and SOX9 (1:500). Thereafter, the cells were washed with PBS and incubated with Alexa Fluor 488 conjugated or Alexa Fluor 594 conjugated secondary antibodies (1:500) in the dark for 1 h at room temperature. Finally, the nuclei were stained with DAPI.

### 2.8. Western Blot

Total proteins were isolated from cultured chondrocytes by using RIPA lysis buffer containing a phosphatase and protease inhibitor cocktail. The samples were incubated on ice for 30 min and centrifuged at 13,000 rpm for 15 min at 4 °C. The protein concentration was determined using the BCA protein assay kit (GeneMark, Atlanta, GA, USA). A total of 30 μg of protein was loaded onto sodium dodecyl sulfate-polyacrylamide gels (SDS-PAGE) and separated through electrophoresis. Subsequently, the protein was transferred to PVDF membranes (PALL, New York, NY, USA). Following the transfer, the membranes were blocked with 5% nonfat milk for 1 h and incubated overnight with primary antibodies against COL I (1:1000), COL II (1:500), COL X (1:500), SOX9 (1:1000) and GADPH (1:10,000) at 4 °C. Thereafter, the membranes were incubated with enzyme-linked secondary antibodies for 1 h at room temperature. To visualize the immunoblots, the enhanced chemiluminescence solution was used according to the manufacturer’s instructions. The protein levels of COL I, II, X and SOX9 were compared between the treatment and control groups (GAPDH) through a semiquantitative intensity analysis (normalized by the respective α-tubulin and background) with ImageJ software.

### 2.9. Statistical Analysis

All quantitative data were expressed as the mean ± standard deviation. Significant differences between the experimental and control groups were examined using one-way analysis of variance. All analyses were performed using IBM SPSS Statistics for Windows, version 22 (IBM Corp., Armonk, NY, USA). A *p*-value of <0.05 was considered statistically significant.

## 3. Results and Discussion

### 3.1. Cell Culture

First, we examined primary (P0) versus passaged (P1–P4) articular chondrocytes obtained from rabbit articular joints. These chondrocytes cultured in the 2D flask served as the conventional model for examining the mechanisms of dedifferentiation or redifferentiation associated with the chondrocyte phenotype treated with chemicals in the present study. Figure 1A presents the morphology of freshly isolated (P0) and passaged P1–P4 chondrocytes. The P0–P1 chondrocytes were small, exhibited fibroblast-like sharp spindles and had no obvious morphological changes. When the chondrocytes were subcultured to the P2, P3 and P4 passages, they elongated into a fibroblast-like structure and reached confluency in a monolayer manner without aggregate formation. An increasing number of chondrocytes exhibited a polygonal shape with large and flattening characteristics after passage through three or four cultures. Moreover, the cell area increased significantly after passage through three or four cultures (Figure 1B). The Western blot analysis indicated that the proportion of chondrocytes producing COL II and SOX9 decreased with the increased passage of in vitro expanded 2D cultures. SOX9 is considered a master transcription factor for chondrocyte differentiation; it drives changes in the expression of ACAN, COL II and COL I [23]. The SOX9 protein could be readily identified through Western blot (Figure 1C). P3 chondrocytes displayed a pronounced enhancement of the SOX9 protein; however, the SOX9 protein substantially decreased with the increased passage of the chondrocyte culture. By contrast, hypertrophic chondrocytes with a large cell area (Figure 1B) exhibited significantly increased COL X production after passage through three cultures (Figure 1C). The result was consistent with other studies, which indicated that the chondrocytes underwent hypertrophy and deposited COL X in a long-term culture period [24,25]. Conversely, Western blot revealed that chondrocytes expressed less COL I protein, which might be because freshly purified chondrocytes maintain their phenotype until passage 4 and secrete less COL I, which could not be detected through Western blot.

The collagen and GAG ELISA assay analyses indicated that the chondrogenic expression of COL II and GAG decreased with the increased passage of in vitro expanded chondrocytes (Figure 1D,E). Reports have indicated that transcription factor SOX9 is associated with chondrocyte differentiation and COL II synthesis induction. The co-expression of the transcription factor SOX9 with COL II and aggrecan was linked and suggested to play a major role in chondrogenic differentiation, cartilage homeostasis and repair at the early stages of OA [26]. Furthermore, immunofluorescence-positive staining demonstrated that COL II and SOX9 staining simultaneously decreased after successive multiple subcultures (Figure 2A). The semiquantitative areas of positive staining determined by ImageJ software for COL II (Figure 2B) and SOX9 (Figure 2C) were significantly decreased as compared with P2 chondrocytes. All findings demonstrated that the chondrocytes undergo dedifferentiation during passage through cultures. However, P2 chondrocytes exhibited an enhanced expression of SOX9 and COL II. We believe that, in the early stages of osteoarthritis, injured cartilage still strives to repair and maintain hemostasis, just like passage 2 chondrocytes in this presenting study, which still can enhance the expression of SOX9 and COL II. On the basis of the high expression and maintenance of the chondrogenic phenotype in P2 chondrocytes, the effect of farnesol on IL-1β-stimulated chondrocyte differentiation was evaluated in P2 chondrocytes.

### 3.2. Effect of Farnesol and Farn/HA Nanoparticle on Cell Viability

The in vitro effect of farnesol and a Farn/HA nanoparticle on the cell viability was examined through a MTT assay (Figure 3A). No significant reduction in the viability of the chondrocytes was detected after 24 h of incubation with <0.4-mM farnesol. The half-maximum inhibitory concentration (IC_50_) of farnesol on the chondrocytes was approximately 0.45 mM. Transmission electron microscopy revealed that the fabricated Farn/HA nanoparticle had a mean diameter of approximately 58.0 ± 10.4 nm (Figure 3B). The EE of farnesol in the Farn/HA nanoparticle was approximately 90%, and approximately 90% farnesol was released after incubation for 10 h in a phosphate-buffered solution (Figure 3C). The cell viability of the chondrocytes was remarkably inhibited after the addition of the 4-mM Farn/HA nanoparticle (Figure 3D). The IC_50_ of the Farn/HA nanoparticle on the chondrocytes was approximately 1.9 mM (4.2-fold increase as compared with pure farnesol). In our previous study, we demonstrated that the encapsulation of compounds and drugs by using HA nanoparticles could effectively decrease the cell viability inhibitory effect of compounds and drugs in cells; moreover, with HA nanoparticles, a greater proportion of the drug could be encapsulated, thus enhancing the therapeutic effect through a slowed drug release [22].

### 3.3. Effect of Farnesol on IL-1β-Stimulated Dedifferentiated Chondrocytes

IL-1β is a pleiotropic proinflammatory cytokine that plays a role in mediating cartilage degradation in osteoarticular diseases. Reports have indicated that IL-1β exacerbates local tissue inflammation through the stimulation of chondrocytes to activate matrix-degrading enzymes, such as MMPs, which downregulate the expression of matrix components [27,28]. Furthermore, the expression of inducible NOS (iNOS) was upregulated in chondrocytes, leading to the excessive release of other catabolic inflammatory cytokines [29]. In the current study, 10-ng/mL IL-1β was added to the P2 chondrocyte cultures and cultured for 24 h; subsequently, the IL-1β-stimulated chondrocytes were treated with 0.4-mM pure farnesol, a Farn/HA nanoparticle (the final concentration of farnesol was 0.8 mM, denoted as Nano-Farn 0.8 mM) and a pure HA nanoparticle. The production of COL I, COL II, GAG and PGE2 synthesis was analyzed using the ELISA kit and immunofluorescence assays. P2 chondrocytes stimulated with IL-1β exhibited a significantly lower production of COL II and increased production of COL I as compared with that in the control group (P2 chondrocytes not stimulated with IL-1β) after 1-, 3- and 5-d culturing (Figure 4A,B). Furthermore, the synthesis of the inflammatory mediator PGE2 significantly increased after stimulation with IL-1β (Figure 4C). The current findings demonstrated that IL-1β downregulates chondrocytes to express matrix components and decrease COL II production. In addition, expression of the inflammatory mediators PGE2 and COL I increased significantly after the stimulation of chondrocytes with IL-1β, indicating that IL-1β led to their inflammation and dedifferentiation.

The main goal of this study was to evaluate the restoration effect of farnesol, particularly in terms of the expression of COL II and COL I proteins and GAG synthesis, on dedifferentiated chondrocytes. COL II production was significantly higher than the IL-1β-stimulated group after treatment with farnesol and the Farn/HA nanoparticle (Figure 4A). By contrast, COL I production was suppressed and significantly decreased to much lower levels than the IL-1β-stimulated group (Figure 4B). The results demonstrated that farnesol could restore the production of chondrogenic COL II and suppress the production of COL I on the IL-1β-stimulated dedifferentiated chondrocytes. The pure nanoparticle exerted a similar but somewhat milder effect on the farnesol-treated group. Furthermore, slow-release farnesol from HA-encapsulated farnesol nanoparticles exhibited a superior promotion on COL II production and remarkable inhibition of COL I production with 3- and 5-d culturing periods. HA is an important constituent of ECM that can bind to ECM molecules and cell surface receptors, thereby regulating cellular behaviors such as proliferation, migration, development, recognition and morphogenesis and some physiological functions. By having unique properties and being a constituent of GAG and articular cartilage, HA would exert cellular effects on the dedifferentiated chondrocytes. However, we focused the restoration effect from the farnesol component and just used HA nanoparticles as a compare group. In fact, the potential effect of HA nanoparticles will be examined in detail in the animal studies for their lubrication, cellular effects and slow-releasing farnesol from Farn/HA nanoparticles.

PGE2 is involved in inflammation and the symptoms of OA and modulates PGE2-derived signaling to play a role in chondrocyte metabolism, affecting the structure of ECM [30]. Treatment with farnesol led to a significant decrease in PGE2 secretion to a similar level as in the control group (Figure 4C). A decrease in PGE2 indicates that farnesol suppresses the inflammation response in IL-1β-stimulated chondrocytes and is probably beneficial in the restoration of chondrocytes and their functions. Effectively bringing down the PGE2 level can improve clinical symptoms in addition to restoring cartilage function. Consistent with our previous published data, farnesol effectively suppressed the inflammation response in different cell lineages [31]. Furthermore, immunofluorescence staining demonstrated almost no positive staining for both COL II and SOX9 after stimulation with IL-1β in comparison with the control group (Figure 5A). The semiquantitative areas of positive staining determined by ImageJ software for COL II and SOX9 analysis in farnesol-treated groups were significantly larger than those for the IL-β-stimulated group (Figure 5B,C). These results suggest that farnesol could restore the ability of IL-1β-stimulated dedifferentiated chondrocytes to secrete COL II and SOX9 and further prevent their dedifferentiation in vitro.

### 3.4. Effect of Farnesol on the GAG Level in IL-1β-Stimulated Dedifferentiated Chondrocytes

Chondrocyte dedifferentiation is defined as the gradual loss of ECM components, such as GAG, COL II and ACAN; simultaneously, the cell shape changes and is characterized by the expression of COL I or COL X [32]. Therefore, alternations in GAG have been associated with the degeneration of articular cartilage, as well as the maintenance of appropriate cell structures and ECM component concentrations, which are crucial for articular cartilage. A significant decrease was observed in the GAG concentration of IL-1β-stimulated chondrocytes, indicating that chondrocytes undergo inflammation and dedifferentiation, gradually losing their ability to secrete GAG after stimulation with IL-1β (Figure 6A). In comparison with the IL-1β-stimulated group, the production of GAG could be regained and increased in the presence of 0.4-mM farnesol or the 0.8-mM Farn/HA nanoparticle. Farnesol is a sesquiterpene alcohol with anti-inflammatory and antioxidant properties. Our previous studies have revealed that farnesol has a significant inhibitory effect on inflammatory factors (such as IL-6 and TNF-α) in third-degree burns and ultraviolet B-induced sunburns in rat studies [17]. Farnesol remarkably and effectively alleviates the inflammatory response caused by skin damage. Consistent with the published data, farnesol was observed to exert an anti-inflammation effect on IL-1β-stimulated chondrocytes. Moreover, immunofluorescence staining demonstrated a significantly increased positive staining for AGG after the IL-1β-stimulated cells were treated with farnesol (Figure 6B). These results indicated that farnesol could suppress the inflammation caused by IL-1β and restore the concentrations of the ECM components in chondrocytes.

### 3.5. Effect of Farnesol on the Dedifferentiated and Inflammatory Proteins Levels in IL-1β-Stimulated Dedifferentiated Chondrocytes

We explored the effect of farnesol treatment on IL-1β-stimulated chondrocytes through the quantification of dedifferentiation according to the COL I, COL X, MMP-1, iNOS and IL-6 inflammatory protein levels through Western blotting (Figure 7). We observed that these protein levels increased in IL-1β-stimulated chondrocytes as compared with the controls. These dedifferentiation and inflammation proteins decreased to a level lower than that in the control group after treatment with farnesol. These results are comparable with previous findings [33]. In IL-1β-stimulated chondrocytes, iNOS-catalyzed NO production led to increased MMP-1 expression, and similarly, IL-6 triggered the synovial membrane to produce MMP-1 and -13, degrading the cartilage structure and COL II. Reports have indicated that IL-6 plays a crucial role in cartilage degeneration through MMP induction in the joint [34,35]. Taken together, the stimulation of chondrocytes with IL-1β led to inflammation in chondrocytes, leading to inflammatory cytokine production and the secretion of dedifferentiated COL I and COL X proteins. All changes caused by IL-1β stimulation on chondrocytes accelerated cartilage degeneration and OA progression.

### 3.6. Restoration Effect of Farnesol in Terms of COL II and GAG Levels in P4 Chondrocytes

In articular tissue engineering, a long expansion time and multiple passaging of chondrocytes are generally necessary to acquire enough chondrocytes. In the present study, we found that P4 chondrocytes lost their chondrogenic phenotype and their ability to secrete COL II and GAG (Figure 1), gradually exhibiting dedifferentiation characteristics. To delay the dedifferentiation progress and to potentially restore the chondrogenic functions of multiple-passaged chondrocytes, we further tested the restoration effect of farnesol through the quantification of COL II and GAG by using ELISA kits. Treatment with pure farnesol or HA-encapsulated farnesol after 1-, 3- and 5-d culturing of P4 chondrocytes restored their function, increasing COL II production and GAG synthesis (Figure 8). In particular, GAG synthesis drastically increased after treatment with farnesol (Table 1).

Our published papers have revealed that farnesol exerts remarkable and effective alleviation effects on inflammation (based on IL-6 and TNF-α inflammatory surveys) in third-degree burns and ultraviolet B-induced sunburns in a rat model. Furthermore, farnesol promotes the production of collagen and facilitates the healing of injured skin. In another paper, we reported that farnesol could modulate the levels of IL-6 and TNF-α in myoblasts and tenocytes and further enhance collagen production in rotator cuff injuries to accelerate healing [21]. The results demonstrated a superior repair in rotator cuff injury with farnesol than with the commercially available product. In particular, slow-release farnesol from the prepared membrane resulted in the marked healing of rotator cuff tear in a shorter implantation period. To our knowledge, this was the first trial using farnesol, an organic anti-inflammatory agent that promotes collagen production, to accelerate rotator cuff repair. On the basis of this experience, we intend to explore the effects of sesquiterpene farnesol, which has anti-inflammatory effects and enhances collagen production, on the maintenance or restoration of the chondrocyte phenotype in dedifferentiated chondrocytes.

IL-1β plays an essential role in stimulating the production of involved inflammatory mediators, such as MMPs, to accelerate cartilage degradation or damage [36]. Many studies have indicated that IL-1β stimulation can activate NF-kB to regulate the production of mediators NO and PGE2, which, in turn, activates MMP-1 and -13 secretion and controls cellular inflammation, ECM synthesis and apoptosis [37]. Meanwhile, IL-1β inhibits the synthesis of proteoglycans and collagen, all of the above contributing to joint degeneration. The present study revealed that, (1) in response to 10 ng/mL of IL-1β, chondrocytes exhibited dedifferentiation characterized by the increased production of COL I, COL X, PGE2, iNOS and MMP-1 and decreased production of COL II, SOX9 and GAG, and (2) farnesol treatment relieved the IL-1β inflammation effect and restored the production of COL II, SOX9 and GAG. By contrast, the production of COL I, COL X and PGE2 was equally suppressed in the control group (Figure 9). Furthermore, reports have indicated that the addition of HA into biomaterial scaffolds significantly accelerates the early-stage gene expression of SOX9 and COL II [38,39]. Thus, we used HA as a carrier material to encapsulate farnesol and found that farnesol encapsulated inside HA nanoparticles exerts superior anti-inflammation and restoration effects in a long-term culture period, which is attributed to the slow release of farnesol. The results suggest that farnesol exerts chondroprotective effects on IL-1β-stimulated dedifferentiated chondrocytes and may delay the occurrence or progression of OA [40]. However, the mechanism by which farnesol induces the secretion of COL II and COL I in IL-1β-stimulated chondrocytes is unclear, and whether it involves TGF-β or BMP-2 receptors merits further study (Figure 9). Crucially, whether farnesol can prevent OA progression or act as a potential therapeutic agent in OA treatment will be evaluated in a rabbit model.

## 4. Conclusions

In the present study, farnesol treatment of IL-1β-induced dedifferentiated chondrocytes significantly restored the production of ECM COL II and GAG, suggesting that farnesol could maintain and restore the phenotypes of chondrocytes and prevent their dedifferentiation in vitro. The therapeutic effects of farnesol and the HA-encapsulated farnesol nanoparticle should be investigated in a rabbit model with the degeneration of articular cartilage or OA through intraarticular injection, with specific consideration toward the hypothesis of redifferentiation of dedifferentiated chondrocytes.

## Figures and Tables

**Figure 1 pharmaceutics-14-00186-f001:**
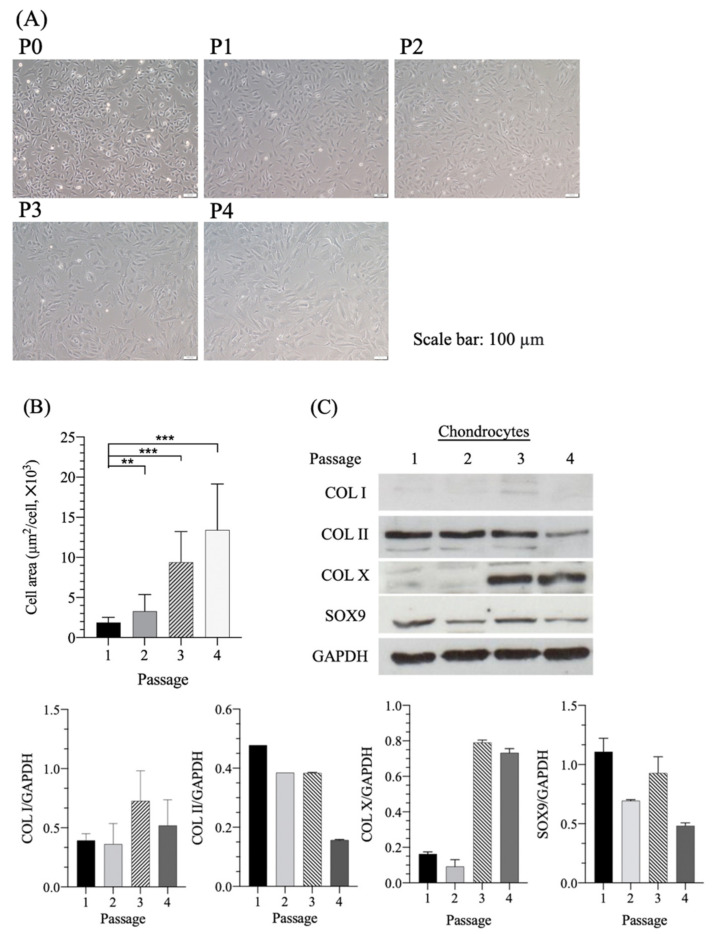
(**A**) Morphology of P0–4 chondrocytes and (**B**) cell area of P1–4 chondrocytes (after 2-d culturing). (**C**) Western blot assay of P1–4 chondrocytes and semiquantitative analysis of COL I, COL II and COL X and SOX9 versus GAPDH. Enzyme-linked immunoassay (ELISA) kit assays of (**D**) COL II and (**E**) GAG concentrations of P1–4 chondrocytes. * *p* < 0.05, ** *p* < 0.01 and *** *p* < 0.001 (*n* = 3).

**Figure 2 pharmaceutics-14-00186-f002:**
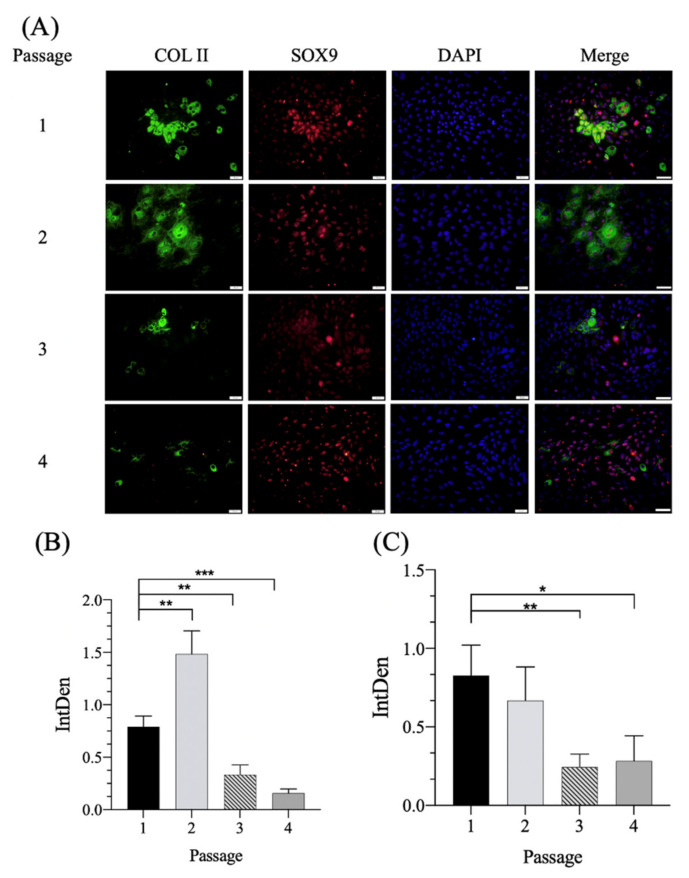
(**A**) Immunofluorescence staining of P1–4 chondrocytes. Semiquantitative assay through immunofluorescence staining determined using ImageJ software of (**B**) COL II and (**C**) SOX9. Green color: COL II staining, red color: SOX9 staining and blue color: DAPI staining. Scale bar: 50 µm. * *p* < 0.05, ** *p* < 0.01 and *** *p* < 0.001 (*n* = 4).

**Figure 3 pharmaceutics-14-00186-f003:**
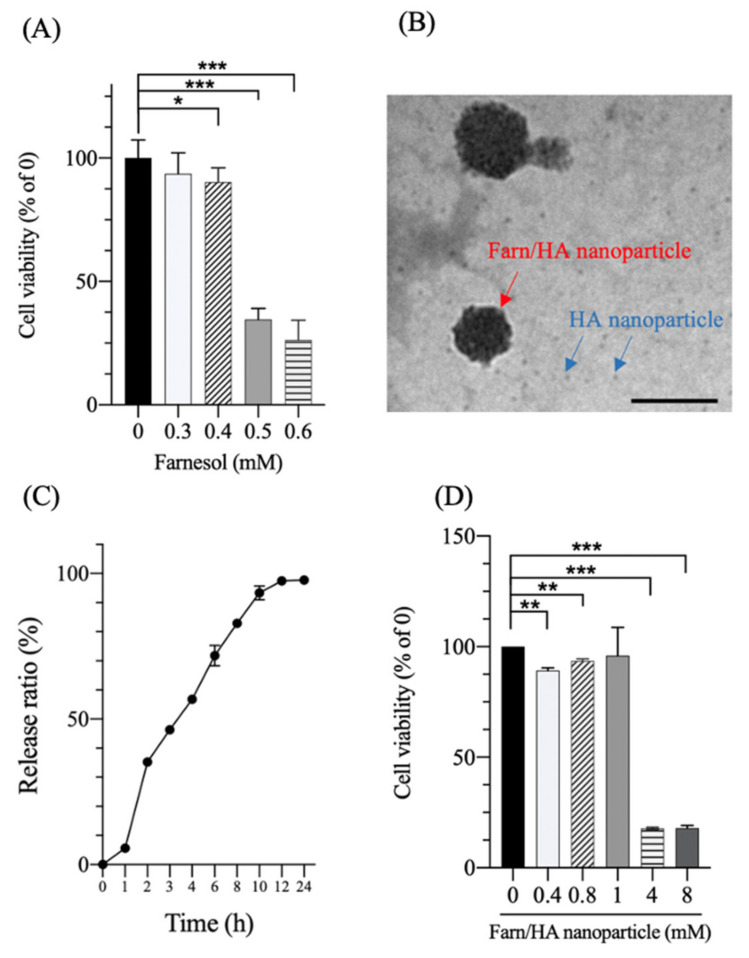
(**A**) Cell viability based on the 3-4,5-dimethylthiazol-2-yl-2,5-diphenyltetrazolium bromide (MTT) assay of farnesol. (**B**) Transmission electron microscopy image of the hyaluronan-encapsulated farnesol (Farn/HA) nanoparticle. The encapsulation of farnesol was approximately 90% (0.8-mM farnesol was added to produce the hyaluronan (HA) nanoparticle). The Farn/HA nanoparticle was 58 ± 10.4 nm in diameter. Scale bar: 100 nm. (**C**) The release profile of farnesol. Approximately 90% of farnesol was released after a 10-h culture. (**D**) Cell viability based on the MTT assay of the Farn/HA nanoparticle. * *p* < 0.05, ** *p* < 0.01 and *** *p* < 0.001 (*n* = 3).

**Figure 4 pharmaceutics-14-00186-f004:**
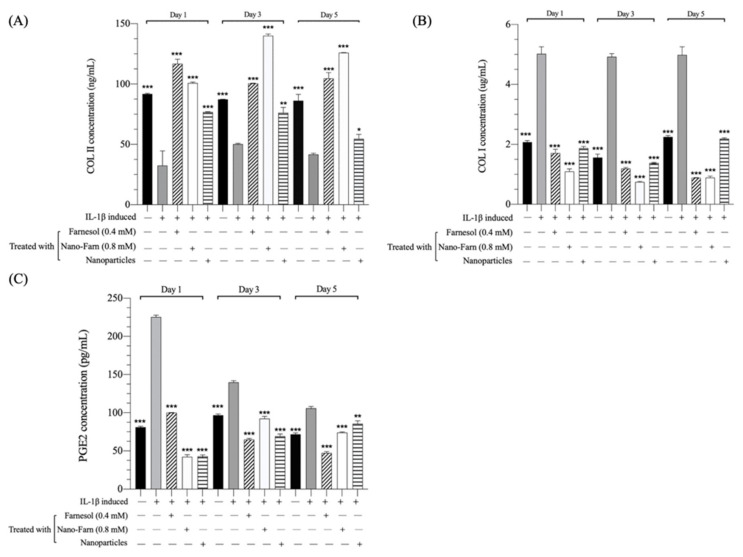
Enzyme-linked immunoassay (ELISA) kit assays of production of (**A**) COL II and (**B**) COL I proteins. (**C**) PGE2 synthesis of IL-1β-induced dedifferentiated chondrocytes treated with farnesol and the Farn/HA nanoparticle after 1-, 3- and 5-d culturing as compared with IL-1β-induced dedifferentiated chondrocytes. * *p* < 0.05, ** *p* < 0.01 and *** *p* < 0.001 (*n* = 3). +: treated, −: untreated.

**Figure 5 pharmaceutics-14-00186-f005:**
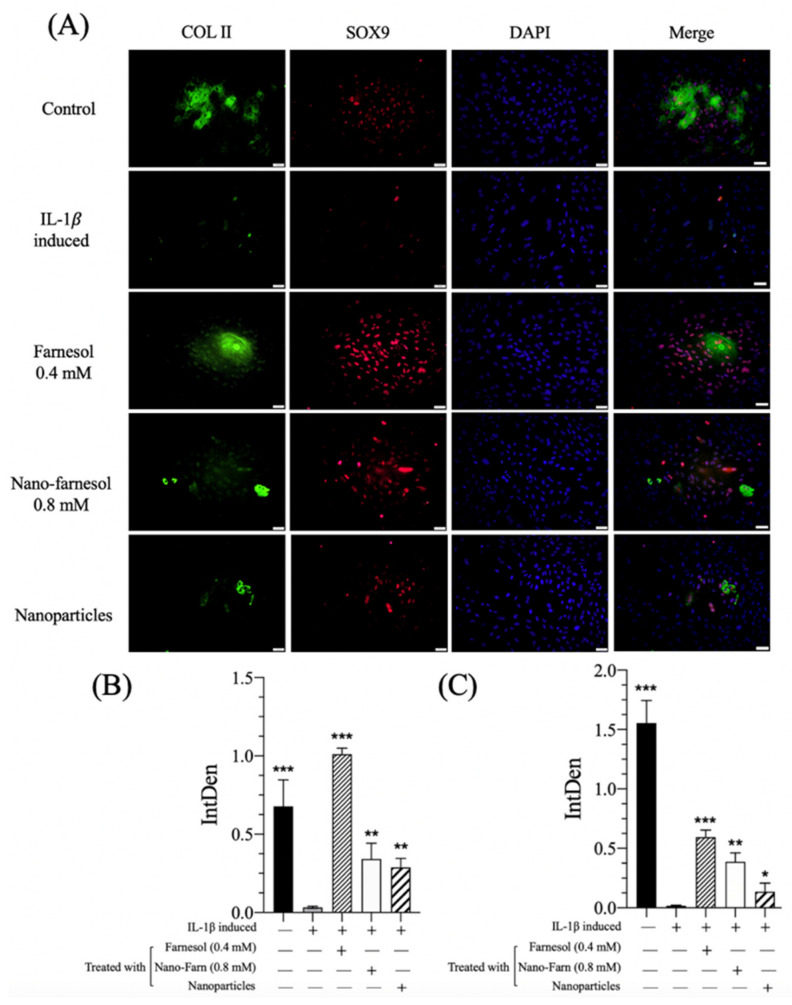
(**A**) Immunofluorescence staining and a semiquantitative assay of (**B**) COL II and (**C**) SOX9 of IL-1β-induced dedifferentiated chondrocytes treated with Farn 0.4 mM, Nano-Farn 0.8 mM and HA nanoparticles (after 5-d culturing). Green: COL II staining, red: SOX9 staining and blue: DAPI staining as compared with IL-1β-induced dedifferentiated chondrocytes. * *p* < 0.05, ** *p* < 0.01 and *** *p* < 0.001 (*n* = 3). Scale bar: 50 µm. +: treated, −: untreated.

**Figure 6 pharmaceutics-14-00186-f006:**
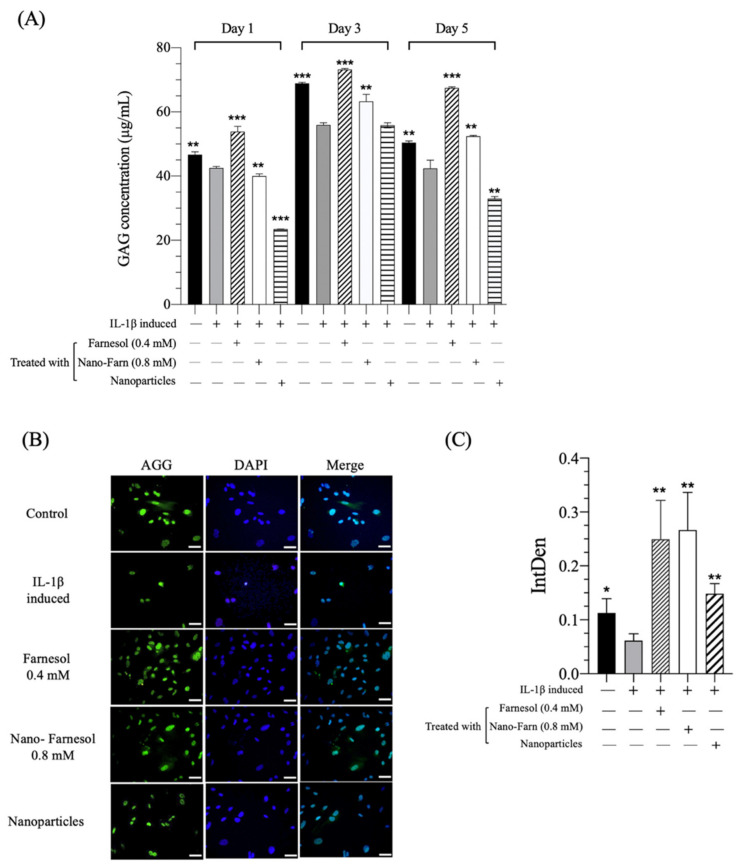
(**A**) GAG synthesis of IL-1β-induced dedifferentiated P2 chondrocytes after 1-, 3- and 5-d culturing determined through the Blyscan Glycosaminoglycan Assay kit. (**B**) Immunofluorescence staining and (**C**) Semiquantitative assay of AGG of IL-1β-induced dedifferentiated chondrocytes treated with Farn 0.4 mM, Nano-Farn 0.8 mM and HA nanoparticles (after 5-d culturing). Green: GAG staining and blue: DAPI staining as compared with IL-1β-induced dedifferentiated chondrocytes. * *p* < 0.05, ** *p* < 0.01 and *** *p* < 0.001 (*n* = 3). Scale bar: 50 µm. +: treated, −: untreated.

**Figure 7 pharmaceutics-14-00186-f007:**
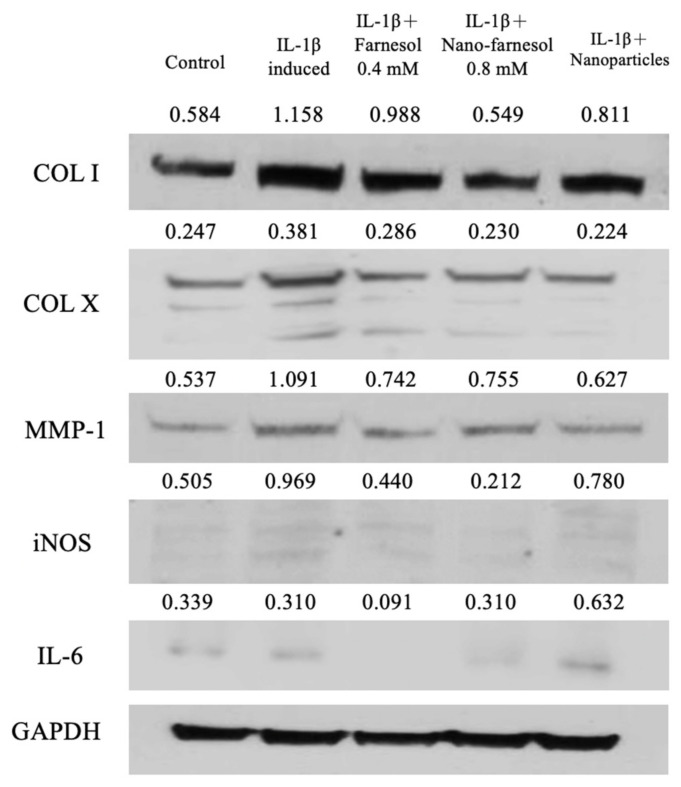
Western blotting of dedifferentiation proteins COL I and COL X and inflammation proteins MMP-1, inducible NOS (iNOS) and IL-6. The values were quantified using ImageJ software. GAPDH was used as the loading control.

**Figure 8 pharmaceutics-14-00186-f008:**
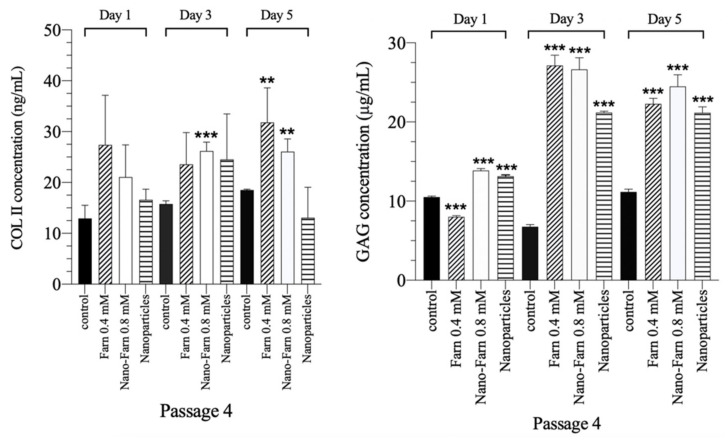
Effect of farnesol, Nano-Farn and nanoparticle treatments on P4 chondrocytes in terms of COL II production and GAG synthesis assayed using the ELISA kit after 1-, 3- and 5-d culturing. ** *p* < 0.01 and *** *p* < 0.001 (*n* = 3) as compared with the control group (w/o treatment).

**Figure 9 pharmaceutics-14-00186-f009:**
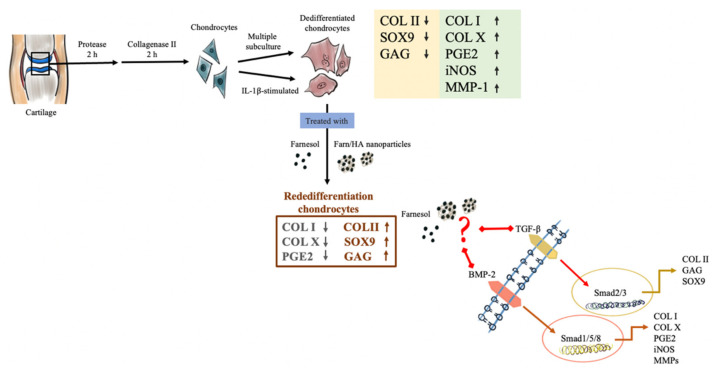
Scheme of the dedifferentiation and restoration of chondrocytes after farnesol treatment.

**Table 1 pharmaceutics-14-00186-t001:** Increase in COL II and GAG in P4 chondrocytes treated with farnesol, Farn/HA and nanoparticles as compared with the control group (from Figure 6). ** *p* < 0.01 and *** *p* < 0.001 (*n* = 3).

	1-d			3-d			5-d		
**COL II**	Farn 0.4 mM	Farn/HA 0.8 mM	Nanoparticles	Farn 0.4 mM	Farn/HA 0.8 mM	Nanoparticles	Farn 0.4 mM	Farn/HA 0.8 mM	Nanoparticles
	2.27 ± 1.22	1.74 ± 0.84	1.33 ± 0.43	1.48 ± 0.33	1.65 ± 0.04 ***	1.54 ± 0.51	1.72 ± 0.38 **	1.4 ± 0.15 **	0.70 ± 0.31
**GAG**	Farn 0.4 mM	Farn/HA 0.8 mM	Nanoparticles	Farn 0.4 mM	Farn/HA 0.8 mM	Nanoparticles	Farn 0.4 mM	Farn/HA 0.8 mM	Nanoparticles
	0.76 ± 0.08 ***	1.32 ± 0.03 ***	1.24 ± 0.01 ***	4.01 ± 0.16 ***	3.94 ± 0.24 ***	3.13 ± 0.15 ***	1.99 ± 0.05 ***	2.19 ± 0.12 ***	1.89 ± 0.04 ***

## Data Availability

The data presented in this study are available in the paper.

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
