# Peer review of "Restoration of the Phenotype of Dedifferentiated Rabbit Chondrocytes by Sesquiterpene Farnesol"

_pharmaceutics, 2022, doi:10.3390/pharmaceutics14010186_

Round 1
Reviewer 1 Report
The authors investigated the restoration effects of farnesol on dedifferentiated chondrocytes.
The manuscript is clear, precise, easy to understand, and offering potentially important information. The following concerns should be addressed before the manuscript can be considered for publication.
Abstract:
- Please rewrite the following phrase: “Chondrocytes dedifferentiated through continued subculture or inflammation lose their cartilage phenotype, and subsequently synthesizing collagen types I and X (COL I and 15 COL X).”
Introduction:
- Line 39-42 - Please add some references.
- Line 50-54 - Please, rewrite this sentence.
- Line 61-63 -Please add more references.
- Line 86-88. Please add more references.
- Please, explain better how farnesol can be included in Tissue Engineering strategies.
Materials and Methods:
- Were the chondrocytes characterized?
- Please indicate which concentrations of farnesol or Farn/HA nanoparticles were used.
Results:
- Figure 1 should be divided into different figures.
- Figure 1 B – Was this measured after how many days?
- How do the authors know that at passage 4, the cells are dedifferentiated chondrocytes and not a more pure population of chondrocytes? When cells are first isolated, other cell populations can be found at P0 -P1.
- Line 248- The authors should not state anything regarding proliferation of cells at P2, since they did not perform any proliferation assay.
- Figure 1 A legend – Passage 0 was also observed. Please, include in the legend.
- Transmission electron microscopy should be added in the materials and methods section.
- Line 303 – why did the authors chose a concentration of 10 ng/mL of IL-1b? Why did the authors chose P2 chondrocytes?
- Line 324 – The authors should explain better the results of pure nanoparticle compared to the farnesol-treated group.
- Semiquantitative assay through immunofluorescence staining do not give appropriate conclusions about these results. The authors should include gene expression analysis, such as qRT-PCR assay.
- Line 358.360 – Please, include some references, such as:
- Carvalho MS, Silva JC, Hoff CM, Cabral JMS, Linhardt RJ, da Silva CL, Vashishth D. Loss and rescue of osteocalcin and osteopontin modulate osteogenic and angiogenic features of mesenchymal stem/stromal cells. J Cell Physiol. 2020 Oct;235(10):7496-7515. doi: 10.1002/jcp.29653.
Author Response
Please refer the attached file

Reviewer 2 Report
The study is devoted to the evaluation of sesquiterpene Farnesol effect on inflammation regulation and phenotype stability of cultured rabbit chondrocytes in in vitro model. The work continues the ongoing research of the authors on its anti-inflammatory and healing mechanisms for different pathologies. The purposes are clearly stated and the methodology is well-designed. The provided results seem quite reliable. However, the manuscript itself seems to be unfinished and roughly prepared. All the sections in the end of the manuscript concerning conflict of interest, funding, etc. should be fulfilled with care before the manuscript is sent for reconsideration. All the technical template text should be replaced or deleted.
The Discussion section is missing, although the last two paragraphs of the Results sec. look much like unfinished Discussion. Il-1β function description is repeated throughout the text several times, it should be left in Introduction and Discussion. Several figures are overwhelmed with information and have to be optimized. Additionally, sometimes it is hard to draw from the fig. caption to which particular experiment the given results belong. In particular:
Figure 1 contains too much elements and must be divided to separate figures for better perception. Group comparison reliability marked with various * symbols must be explained in figure caption. Explanation of the results should be given in the main text, not in the caption. The association of (G) and (H) elements with immunofluorescence experiment should be clearly stated in fig. caption.
Figure 3 is also suggested to be divided for two separate figures for better perception.
Fig.3.(A) – (C): the particular measurement technique the shown results originated from should be clearly stated in figure caption. The same for Figure 4A.
In Methods section characterization of Farn/HA nanoparticles by TEM technique must be given.
Lines 105-106: the country origin of the reagents must be given.
Lines 244-245: “Consistent with other studies, the expressions of SOX9 and COL II in this study simultaneously decreased during passage through chondrocyte cultures” – the particular previous studies consistent with the result must be given here.
Line 393: “These results are comparable with previous findings”. The citation of previous research must be given.
Line 396: extra closing parenthesis.
Section 2.3 is given in bold. Why?
It’s a bit confusing how pË‚0.001 reliability can be obtained from 3 measurements. The authors should provide the results of individual measurements in supplemental materials.
Figure 5. There is no need to repeat the details of the experimental procedure in fig. caption. The mention of particular experiment from Materials sec. is quite enough.
English should undergo corrections throughout the text. For example, “effect at” should be changed to “effect on” in sec. 3.4 and 3.5 headings; “Schematic” in Fig.7 caption should be changed to either “schematic view” or “scheme”, etc.
Questions for the authors concerning the main concept:
- In some reported measurements the effect of HA nanoparticles is almost as pronounced as that of Farnesol-containing drugs (for example, fig.6). However, it is small attention paid to the fact in the discussion. What can be the mechanism of HA effect? Can it be considered as a separate treatment?
- Possible mechanism of Farnesol action presented in Fig.7 need to be described in the text in more detail. The explanation of the lower part of Fig.7 containing TGF- β and BMP-2 ways is missing, and from the fig. it is not quite clear what possible effects are supposed by the authors.
- The inflammation suppression and matrix synthesis stimulation actions of Farnesol are mentioned in the same time (lines 419-426). Yet, the results of the present study can directly support only the first way. What arguments can be provided in support of the stimulation effect of the used drugs?
Author Response
Please refer the attached file

Reviewer 3 Report
General comment:
In this manuscript the authors analysed the effect of farnesol on the restoration of the phenotype of dedifferentiated Rabbit Chondrocyte. The concept is interesting, and the strategy has the potential to be applied in many tissues engineering approaches. The authors focused the analysis of the effect of Farnesol on a chondrocyte rabbit model and shows the effect of farnesol when encapsulated or not in Hyaluronan nanoparticles on the dedifferentiation of chondrocytes due to passaging and on the restoration after inflammatory response triggering with IL-1β. Despite the results are of interest, some of the data will benefit from the inclusion of additional controls. A detailed discussion the results obtained and of the aim of the proposed strategy in the tissue engineering field is also recommended. Please find below my comments and suggestions.
Abstract:
- Line 12: I suggest amending the term “destruction” with “progressive degeneration”.
- Line 14: The authors should underline the link between osteoarthritis and the dedifferentiation of chondrocytes in culture.
Materials and Methods:
- Line 103: I suggest adding a paragraph also for this section.
- Line 151: The text of this section appears to be in bold.
Results
Cell Culture - I suggest modifying the title of this paragraph to better reflect the results following the style of the other paragraphs of the results section of the manuscript.
- Figure 1: The authors should indicate in the caption what is represented by the error bars and they could show the statistical analysis in all the graphs of figure 1C also if they are not significant (e.g.: n.s.). Moreover, the authors should better clarify in the text and in the caption what figure 1G and 1H are referring to.
- It would be interesting to also evaluate the gene expression of the same chondrogenic markers by analysing the variation in mRNA amount via RT-qPCR (also valid for the farnesol effect).
Effect of Farnesol and Farn/HA Nanoparticle on Cell Viability
- Figure 2B: I suggest indicating more specifically in the material and method section how the quantification of the diameters of the nanoparticles was achieved.
- Figure 2D: I suggest clarifying how the concentration of farnesol was calculated when encapsulated with HA. Is the graph referring to the theoretical encapsulated amount or to the calculated concentration of Farnesol in the media after release?
Effect of Farnesol on IL-1 β Stimulated Dedifferentiated Chondrocytes
- Figure 3 A-B-C: The Authors should indicate in the graphs and in the immunostaining that the farnesol and farnesol/HA treatment were administered after the IL-1β treatment. Moreover, the interpretation of the data will benefit from the addition of a farnesol and farnesol/HA treatment in absence of IL-1 β inflammatory induction.
Effect of Farnesol at the GAG Level in IL-1 β Stimulated Dedifferentiated Chondrocytes - Figure 4A: The Authors should indicate in the graphs and in the immunostaining that the farnesol and farnesol/HA treatment were administered after the IL-1β treatment. Moreover, the interpretation of the data will benefit from the addition of a farnesol and farnesol/HA treatment in absence of IL-1 β inflammatory induction.
Effect of Farnesol at the Dedifferentiated and Inflammatory Proteins Level in IL-1β Stimulated Dedifferentiated Chondrocytes
- The Authors should show the changes in the level of collagen II protein
Restoration Effect of Farnesol in Terms of COL II and GAG Levels in P4 Chondrocytes
- The Authors could evaluate how the farnesol affect the levels of Collagen II and GAG when the Chondrocytes are progressive passaged as indicated in figure 1. This also reflects the need for a detailed explanation of the possible preclinical application of the Farnesol.
-This work will benefit from the confirmation of the data obtain on a human derived chondrocyte cell line, or the choice of the rabbit model should be carefully discussed.
Conclusion:
- I suggest adding a discussion section where the authors should comment on the effect of the farnesol when encapsulated or not, on the choice of the rabbit derived cellular model and on the application of the proposed strategy in the tissue engineering field.

Author Response
Please refer the attached file

Round 2
Reviewer 1 Report
The authors have addressed the reviewer's comments.
Author Response
refer the attached file

Reviewer 2 Report
The main points have been successfully adressed by the authors.
Please, improve the following: Fig.6A, add the method of GAG quantification to the fig.caption.
Author Response
Refer the attached file

Reviewer 3 Report
The authors addressed most of the comments.
However, some key points should be considered:
1- I my interpretation s correct, In almost all the experiment presented, the Farnasol was administered only after the treatment with IL1ß (Fig.4-5-6-7-8). The authours should show also the effect of Farnasol and Farnasol/HA on GAG and Collagen II in a control group where the chondrocytes are not treated with IL1ß.
2- I advice changing the labels also on the graphs of figure 4-5-6-7-8 (not only on the captions) to indicate that the Farnasol experimental groups are treated (or not) with IL1ß.
3- In figure 7 (Western Blot), The authors should show the changes in the level of Collagen II protein togheter with the other markers as well as including a control showing the effect of Farnasol and Farnasol/HA in absence of IL1ß treatment.
Author Response
Refer the attached file
